# The Language of Nature and Artificial Intelligence in Patient Care

**DOI:** 10.3390/ijerph20156499

**Published:** 2023-08-01

**Authors:** Teresa Enríquez, Paloma Alonso-Stuyck, Lourdes Martínez-Villaseñor

**Affiliations:** 1Instituto de Humanidades, Universidad Panamericana, Josemaría Escrivá de Balaguer 101, Aguascalientes 20296, Mexico; 2Facultad de Psicología, Universitat Abat Oliba CEU, Bellesguard 30, 08022 Barcelona, Spain; palonsos895@uao.es; 3Facultad de Ingeniería, Universidad Panamericana, Augusto Rodin 498, Ciudad de México 03920, Mexico; lmartine@up.edu.mx

**Keywords:** language of nature, artificial intelligence, ethics, patient care

## Abstract

Given the development of artificial intelligence (AI) and the conditions of vulnerability of large sectors of the population, the question emerges: what are the ethical limits of technologies in patient care? This paper examines this question in the light of the “language of nature” and of Aristotelian causal analysis, in particular the concept of means and ends. Thus, it is possible to point out the root of the distinction between the identity of the person and the entity of any technology. Nature indicates that the person is always an end in itself. Technology, on the contrary, should only be a means to serve the person. The diversity of their respective natures also explains why their respective agencies enjoy diverse scopes. Technological operations (artificial agency, artificial intelligence) find their meaning in the results obtained through them (*poiesis*). Moreover, the person is capable of actions whose purpose is precisely the action itself (*praxis*), in which personal agency and, ultimately, the person themselves, is irreplaceable. Forgetting the distinction between what, by nature, is an end and what can only be a means is equivalent to losing sight of the instrumental nature of AI and, therefore, its specific meaning: the greatest good of the patient. It is concluded that the language of nature serves as a filter that supports the effective subordination of the use of AI to its specific purpose, the human good. The greatest contribution of this work is to draw attention to the nature of the person and technology, and about their respective agencies. In other words: listening to the language of nature, and attending to the diverse nature of the person and technology, personal agency, and artificial agency.

## 1. Introduction

In the face of demographic change, the United Nations has promoted the current Decade of Healthy Aging. Estimates are that the percentage of people over the age of 60 will double, from 12% in 2015 to 20% of the world population in 2050. In order to reduce inequalities in health, one of its proposals is to provide person-centered integrated care [1].

Artificial intelligence has been transforming the practice of medicine and healthcare [2]. The availability of sensors, mobile communications, cloud computing, and machine learning techniques enable new intelligent applications to provide better medical and healthcare services at lower costs. Intelligent systems for monitoring of clinical activities, ambient assisted living, diagnosis and prognosis of diseases, and treatment optimization and outcome prediction among others are changing clinical practice [3]. Nevertheless, these benefits entail important risks that have to be considered when developing and deploying an intelligent system for healthcare. Privacy risks, the delegation of critical medical decisions to a system, loss of human agency, opacity in the system’s decision making, the possibility of increasing inequities and biases, and the lack or dilution of accountability are issues that concern the community [4].

The concern for the potential loss of cognitive freedom at the expense of the so-called NBIC (nano-bio-info-cogno) [5] has triggered the formulation of neurorights [6], as, when AI systems are deployed in real-world applications [7,8], they generate risks and undermine individual self-determination [9].

The aforementioned research raises many questions. Is it possible that we are trivializing ethics? Should we use all the possibilities offered by technology? [10]. Could AI in health supersede the human will in decision making? [11]. Is it possible to build an ethical AI? [12]. What would be the requirements for AI decisions to be reliable? [13]. How will it be possible to avoid the so-called black box phenomenon in the choice of patients who will receive health resources? [14].

Will human intervention always be necessary at the end of the decision-making process? [15]. Is the creation of new machine learning models enough to blur the distinction between human and artificial intelligence? [16,17].

It may seem that the solution lies in establishing ethical limits to technological development. However, following this path would lead to minimalistic ethics, whereas the language of nature inclines towards fullness, towards maximalist ethics. This is achieved through virtues, and healthy lifestyle habits that enable individuals to be free [18] (L. II, ch. 6, 1106 a 22–23). The challenge is to strike a balance of maximums between the level of technology and ethical good. The need for spaces of reflection is evident in order to make ethically sensitive decisions [19].

To achieve this balance of maximals, we propose analyzing under what conditions the use of AI would be oriented towards the good of the individual or against it. The hypothesis of this article is as follows: the language of nature can shed light on reaching this balance. In the healthcare field, the language of nature emerges as a guide for a good life, and for the art of living. The language of nature indicates the basic principles present in the human heart, those that people from all cultures share [20]. It is about the fundamental orientation towards the good and the capacity to discern it in each situation [21].

Basic goods such as life, bodily integrity, civil rights, and the right to information are some of these fundamental principles [22] (S. Th. I-II, q. 94, a. 2). The language of nature can be recognized in these goods [23], in the sense that humanity recognizes them as rooted in human nature [24].

## 2. Materials and Methods

To achieve the balance of maximums, the teleological perspective was adopted (telos, in Greek, means purpose), which is characteristic of the philosophy of action of authors such as Plato, Aristotle, and Thomas Aquinas [25]. The primacy of the final cause in decision-making allows for the maximization of the possibilities of the material cause [26,27]. Furthermore, the results of the search in PubMed confirmed the relevance of two particularly clarifying concepts in the field of health sciences: *praxis* and *poiesis*. It is significant to find that most of the literature on AI ethics in the healthcare area has been approached from the perspective of *poiesis* [28]. Therefore, it is necessary to draw attention to the value of *praxis*, as is done in this work.

The philosophical approach was complemented by a psychological perspective. A search in Scopus on “ethical limits of technologies” yields 2157 articles. Adding the term “decision making” reduces the result to 237, and, by also adding the expression “human dignity”, the number of articles is reduced to 5. In the same way, a search in WOS regarding the limits posed by the *“language of nature”* yields 105 articles, which, refining with the previous keywords, are reduced to 31. These investigations have been used to understand the language of nature and, thus, achieve a balance of maximums [21].

In addition, different initiatives of ethical guides and good practices for the development and use of AI have been consulted [29].

When we searched in SCOPUS using the keywords “artificial AND intelligence AND patient AND care”, we obtained 75 results. Nevertheless, when we added the keyword “ethics” to this search, there was only 1 result. The work of Rogers et al. [30] presented the ethical evaluation and description of two clinical decision support AI-based applications, and investigates ethical principles for AI-assisted healthcare applied through frameworks. They highlight the need to identify ethical issues across the AI system’s life-cycle.

The results are presented in the following section under 4 headings:The ethical limits examined in light of the language of nature and the universally accepted principles [31].The factual limits through the Aristotelian analysis of action, which recognizes the human agent as the primary efficient cause and technology as the material or instrumental cause.The intrinsically unlimited nature discovered in Aristotelian action (*praxis* in Greek), whose teleological structure (*telos* in Greek meaning purpose) distinguishes it from *peras* (which is limitation). *Praxis* is distinguished from productive activity (*poiesis* in Greek).The relevant guiding ethical principles for intelligent technologies in healthcare.

This detailed analysis will help us understand that treating technology as an end and the person as a means goes against the language of nature. It also distinguishes between actions whose object is the very realization of the action (*praxis*) and those that are merely means to achieve a result (*poiesis*).

## 3. Results

### 3.1. Analysis from the Limits of the Language of Nature

The Latin etymology of the term *precept* or moral limit expresses this illuminating quality: *praecise coeptum* means “to take exactly“, in order to prohibit what, despite appearances, does not contribute to the good of life [22] [In Ps. 18, n. 5]. In that sense, the limits of the language of nature indications are truthful and constant [32], thus avoiding fallacious or interested interpretation [33]. These limits are based on the first principle of “do good and avoid evil”, whose two parts correspond to the principles of “beneficence” and “non-maleficence” of contemporary bioethical and medical principles (namely beneficence, non-maleficence, autonomy, and justice) [31].

Learning to read the language of nature, respecting its grammar, could become a universal guide [23,34], since it illuminates the correct solution depending on the circumstances [35].

Knowing how to make the right decisions is precisely the skill provided by human virtues as a whole [36]. The language of nature opens to an incessant growth of the correct use of freedom, in which the virtues consist. However, if human agency is renounced for the admirable developments of AI, the possibility of growth in virtues would also be lost.

The temptation to renounce personal agency by unloading work on technological instruments is typically human and is recorded in Plato (5th century BC) regarding the invention of writing:

He who has something in writing no longer cares to keep it in memory [37] (Phaedrus, 275 a–b). In the medical field, the warning is similar: will those who use AI for diagnosis continue to exercise their clinical judgment? The challenge now is to make the most of new technologies to enhance, without weakening, the specifically human professional competencies.

In patient care, intelligent systems and assistive robots can be effective in tasks such as distributing pharmacological treatment, monitoring vital signs, drawing patterns of the most effective therapies, etc. Hence, this suggests the specific values of human nursing: treat the patient with dignity, charity, and care. Therefore, in order not to be displaced, the doctor, the nurse, and the caregiver have to establish limits: distinguish the possibilities of these instruments, no matter how sophisticated they are, and recognize the height of the demands of their own human nature.

The work of Floridi et al. [9] (2.1–2.4, pp. 691–993) identifies four opportunities and risks derived from the misuse and overuse of AI.

Who we can become: enabling human self-realisation, without devaluing human abilities;What we can do: enhancing human agency, without removing human responsibility;What we can achieve: increasing societal capabilities, without reducing human control;How we can interact: cultivating societal cohesion, without eroding human self-determination.

These four warnings are based on the different natures of human agency and artificial agency. Artificial agency does not respond as the subject of its acts, because it is not the first principle of its movement, and it is not the main efficient cause, as will be seen in the next section.

The disproportionate use of AI would entail a significant loss, because it would inhibit the agency of the human being [38]. That is why the language of nature indicates the limits in the use of AI: it keeps it in a position of an instrument, in service to personal agency, without granting it the power to replace the human person.

### 3.2. The Position of AI in the Philosophical Analysis of Human Action

To the ethical limits of the language of nature that prevent technology from displacing human agency and, therefore, situate it as a collaborator and not as a substitute for the human being, we must add the factual limits that also come from the nature of technology, which is intrinsically a means and not an end. Considering the nature of the means and the nature of the ends is also a way of listening to the limits of language of nature.

It is significant that, when Aristotle considers the categories of ends and means, he takes medicine as an example of art and health as an example of purpose. In this regard, Thomas Aquinas continues with the breakdown of the doctor’s task:

“As medicine has no limits in restoring health and each of the arts is unlimited in its end, because they want to achieve it to the maximum, but it is not unlimited in what is pertinent to that end” [39] (Aristotle, Politics, L. 1, c.9, 1257b13). “The doctor, in fact, does not set a limit to the restoration of health, rather, he strives to ensure it as perfect as he can; he sets, instead, limits to medicine, and so he does not give what he can, but what is necessary for the restoration of health; exceeding or missing the due proportion would be missing the measure.” [22] (S.Th. II-II, q. 27, a. 6, co.)

In the cited fragments, three basic elements are already registered: health, medicine, and the doctor as examples of three philosophical categories: end, means and agent, where health is the end, medicine is the means, and the doctor is the agent. The point of convergence is the agent, since they intend the end and the means, but in a different way. Indeed, while doctors unrestrictedly seek the health of their patient (“do not set limits”), they restrict, on the other hand, the use of medicine (“set limits”). It is seen, therefore, how the limitation makes sense depending on the purpose.

In order to discover if the purpose is adequate, it is useful to apply the filter of the limits of language of nature, synthesized in these four questions: Is there a proportion between benefits and losses? Is the person being used as a means? Is their role respected? Is the common good sought?

If attention is paid to the structure of human action, it is logical to place the limit in the use of technologies as a means. “To use is to direct what is at our disposal to achieve something” [40] (L. I, c. 4, ML 34, 20).

AI is a powerful technology. Thus, the fullness of its nature lies in contributing to the purpose for which it is intended: the well-being of the individual.

Aristotle classified the causes of natural entities into four: material, formal, efficient (agent), and final (purpose). This classification is exposed in three fragments [41] (L. II, ch. 3, 194b 17–20); and [42] (L. I, ch. 3, 983a 25-983b; L. V, ch. 2, 1013a 20–1013b). Among the causes there is a hierarchy: depending on the purpose, the agent will introduce a certain form into the matter. In the case of medicine, the purpose is healthcare, and materiality is the body of the patient in the first place, and, then, the set of resources that collaborates in its recovery. Technology, as a material instrument, is at the service of the end, which is the integral good of the patient. Disrupting the order of the means ends up damaging the subject and the end of health: the person.

It should be considered in situations of vulnerability, when it is not easy to apply the rules of “informed consent” (Mallardi, 2005), or to balance the patient’s right to information with their decision-making capacity, diminished by the disease [43].

Michel Schooyans [44] warned of the seriousness of the consequences of defining medicine by its material cause, that is, by the use of medical instruments. An executioner could use medical instruments and, not for that reason, carry out the proper purpose of medicine: healthcare [44] (p. 9). If healthcare is not the purpose of an act carried out with medical instruments, then it is not a medical act but of another nature, such as an act of profit. That is what chrematistics consists of, defined by Aristotle as the “acquisitive art for which it seems that there is no limit to wealth and property” [39] (L. 1, ch. 9, 1256b 9). The intention, be it the health of the person or money, reflects the identity of the agent and transforms the action into its root, even though the latter preserves, in its material appearances, the same object. The difference in intentions in the same action is illustrated in a couple of verses by a new Spanish poetess [45] (109). If we transfer THIS question to the scope of our research, we could ask: who is more interested in curing: the one who cures for profit or the one who profits to cure? And in terms of AI, the question would be: do we use AI to care for the patient or do we use the patient to further develop AI? The latter seems to occur when the means to collect personal information and use it indiscriminately for the development of AI systems are not taken into account.

In the use of any technology, its ultimate purpose should be examined. For the so-called “technological imperative”, the limits of technological power are also the limits of duty. Thus, it displaces the ethical principle “do good and avoid evil” with “everything that can be done should be done”. And, thus, whoever operates according to this imperative, takes technology itself as the purpose of technology. This way of proceeding is part of the so-called “technocratic paradigm” [46] (n. 109) or in “technologism that distorts technology and treats it as an end in itself and not as a means” [38] (p. 17). Medicine, from this paradigm, would treat the person more as a means than as an end. The basic distinction between the nature of both agencies is that, while the human being acts according to the four causal senses, the artificial agents do so only in the sense of the material cause, or, at most, as an instrumental cause. The person is in itself the main purpose: they deserve to be loved for their own sake and, therefore, their integral good should be sought without any measure.

### 3.3. AI and Human Agency: Poiesis and Praxis

An Aristotelian distinction, also clarifying, regarding limits is the one between productive movements (*poiesis*) and exclusively human actions (*praxis*) [47].

In this distinction, Aristotle resorts to an example from the health sciences [42] (L. IX, ch. 6, 1048b).

The concepts of *praxis* and *poiesis*, in their original Greek meaning, are used in the healthcare field to examine the professional competencies of patient caregivers [48].

This excerpt from Aristotle distinguishes actions according to their purpose:

*Praxis* refers to the activity whose purpose is the action itself. Thus, the action that is *praxis* achieves its end while it is being executed. Actions such as observing, listening, accompanying, living together, consoling, and all forms of communication are *praxis* because they are actions whose purpose is in the action itself. A doctor who spends time listening to a patient’s concerns and needs is taking *praxis* actions. *Praxis* should be a priority in the doctor–patient relationship and in personalized care.

*Poiesis* refers to productive activity, that is, useful to obtain something. *Poiesis* is oriented towards an end other than itself. Its purpose is something external, produced by that action. For this reason, it is executed only to the extent necessary to achieve this subsequent purpose. *Poiesis*, in the context of healthcare, can be seen in activities related to the creation of instruments, i.e., medical technology. For example, biomedical engineers who design advanced prosthetics or medical diagnostic equipment are performing *poiesis* actions. Their work focuses on the production of something tangible that contributes to the advancement of medicine and improves the quality of life of patients. Artificial intelligence systems emulate the poietic processes that the human being would do. The creation of AI is based on knowledge of human *poiesis*. Aristotle said: “art imitates nature” [49], but we could say: “artificial intelligence imitates *poietic* human intelligence”.

This distinction provides a criterion for delegating productivity (*poiesis*) to AI, but not practicality (*praxis*), because the "praxical" action is itself finality.

Curative medicine is a productive movement *poiesis* insofar as it achieves changes in the organism. In this process, technologies play a significant role, and as we can see from the major advancements in AI that a large number of care tasks can be performed more efficiently by a robot. Productive actions are oriented towards a result that is different from the action itself. The examples have already been mentioned: dispensing pharmacological treatment, monitoring vital signs, extracting patterns from the most effective therapies, and so on. In all of these, AI is much more efficient than a human: better results, faster, and cheaper.

However, in nursing care, we find more examples of *praxis*: accompanying, dialoguing, consoling, listening, comforting, and forms of love that may be considered non-productive but immensely comforting for a patient. Can one be comforted by a robot? Perhaps something more sophisticated than what a stuffed animal offers could be achieved, but the truth is that physical objects do not fulfill human possibilities. In practical actions, the person who cares for the patient is irreplaceable because the end of the action is the action itself. Human companionship is not entirely replaced by entities of a different nature because “one is said to be alone in the garden even though there may be many plants and animals there” [22] (S. Th., I, q. 31, a. 3, ad 1). On the other hand, a human caregiver is open to a myriad of possibilities in intention and realization.

Empirical studies confirm Aristotle’s conceptual analysis: while *praxis* is more common among nurses, *poiesis* is more common among physicians [50]. When a doctor has the ultimate goal of the well-being of their patients, their empathy is capable of creating a shared reality, thus respecting the centrality of the person [51,52]. This intention is evident in both the great masters of antiquity, Hippocrates and Galen, as well as in the Christian benevolence of medieval monks, which has given rise to the principle of beneficence in modern bioethics [53].

Efforts to teach empathy to robots are known, but will it be possible? [54]. Can robots truly replace humans? Some argue that robots still have much to learn in the field of medicine [55], while others highly value the mind-to-mind relationship that the metaverse introduces us to [56]. Despite this, medicine, as the art of healing, is certainly a productive movement. Hence, it is more than that. It has been said that if medicine was solely a process of diagnosis and therapeutic monitoring in a purely physiological realm, perhaps doctors could be replaced by robots. However, the art of medicine, nursing, and hospitality is of a nature superior to producing health or restoring homeostatic balance.

Thus, the purpose of healthcare extends beyond that: towards the patient themselves. In professional domains related to care of individuals, the challenge of maintaining the justified supremacy of *praxis* over *poiesis* has been highlighted [57].

The person is, by nature, an end in themselves, and treating them as such is to follow the limits of the language of nature. However, one must not confuse the comprehensive well-being of the patient with maintaining the appearance of a nonexistent biological life through excessive therapeutic measures. The ethical limit of using technologies, including AI, lies in the realm of means. In contrast, in terms of the ultimate purpose, for the comprehensive good of the person, there are no limits. An influential medieval author dismisses all limits in the realm of the end: “the measure of love is to love without measure” [58].

### 3.4. Pragmatic Guiding Principles for Healthcare Intelligent Technologies

In order to align the development of artificial intelligence systems to an ethical design according to the language of nature, the most important task in this first stage of development should be defining the ethical guiding principles for the intelligent system. For the case of patient healthcare, we propose to reflect on the bioethical and medical principles [31] adapted by Floridi et al. [9], as well as to include well-known agreed principles by the artificial intelligence community, according to Fjeld et al. [29].

We studied these principles and we combined them, as presented in Figure 1. The first row of Figure 1 presents the principles proposed by Floridi et al. [9], and the second row represents the matching most relevant principles for each correspondent principle of the first row. The intention of this matching process was translating Floridi’s principles into artificial intelligence known issues and concerns. In this way, the revision of the main concerns for an intelligent system has clear guidance for impact assessment.

The ethical possible issues for each healthcare application should be described by revising each column of Figure 1 in order to find the most compelling concerns for a certain application in each step of the lifecycle development. These concerns must only be identified, and ethical risks must be listed. This process should help guarantee that human dignity is preserved and human rights are not violated by conducting a thorough impact assessment as part of the ethical design development process.

## 4. Discussion

Recent advances in sensor technology, data analysis, cloud computing, and the availability of huge amounts of health data allow the creation of new artificial intelligence systems that are transforming the way of providing patient healthcare [59]. Nancy Robert presented examples of how artificial intelligence is changing nursing [60]. Nurses are using data in an electronic medical record to follow a patient’s condition over time using the Rothman Index surveillance solution. This tool with new data analysis algorithms will allow nurses to understand a patient’s illness and needs better and spend more time at the bedside. Artificial intelligence tools can relieve nurses from some operative and administrative burdens and enable them to focus on their professional activities. Microsoft and the Cincinnati Children´s Hospital Medical Center (CCH) tested mobile apps to enhance the patient experience with the collaboration of nurses. Social or companion robots are designed to interact with patients and older adults, performing simple nursing tasks such as ambulatory support, vital signs measurement, and medical administration, among others. With this support, nurses can spend more time with patients, i.e., can perform more *praxis*. Hence, despite all these benefits, we must be aware of the ethical problems involved [61]. Transparency, reproducibility, bias, privacy, and, most importantly, accountability and human control are ethical considerations relevant to patient healthcare. Ultimately, nurses are accountable and responsible for nursing practice and the impact on patient health [62].

In order to deal with and discuss these ethical issues, there are approaches that address privacy, explainability, and transparency, bias from the technical point of view [63]. Moreover, in recent scientific literature, there are also abundant practical guidelines to guide the development of AI for the greater good of patients and caregivers. These guidelines can be grouped into the following four sections:


**Understanding**


Promote the enhancement of humanity through technology [64]. AI developments will allow for a reduction in time spent on routine tasks and enable individuals to engage in higher quality care [65]. This care belongs to the realm of *praxis*, where its value lies more in the process than in the outcome [66]. Love is the greatest force that moves the world [67]. And the measure of love is unlimited love.


**Communication and Education**


Collaborate with social agents in communication and education. (a) Raise awareness through narratives present in the media about the need for common limits, as this is an endeavor shared by all social agents involved in healthcare: family, market, state, etc. Media and networks should transmit a truthful narrative about AI [68] due to the impact they have on the AI–person relationship. (b) Teach within families how to interpret the language of nature, and show respect for the individual and their dignity, as children will be the main disseminators and decision-makers [69]. Educational institutions are where precise conclusions are drawn from this language in each professional field.


**Legislation**


Legislate, supervise, and work towards equity. The democratization of data [70] can also contribute to preventing AI from imposing its code of conduct under the guise of digital freedom [71]. The experience of other countries shows that it is not easy to determine who regulates this issue [72]. However, it is clear that there is an urgency to legislate, both by supporting existing initiatives proposed since 1999 [73] and through various proposals: the technocratic oath [74], an AI oversight agency detailing its permissible uses [75], and governmental decision-making processes [76].


**Decision-Making**


The automatic decisions made by an intelligent system must be considered to support human decisions and therefore should always be revised by clinicians or care givers. Humans have to retain absolute power over machines [77]. Human control also entails deciding which tasks and decisions to delegate and which ones to not. In addition, human control is linked with accountability and professional responsibility. It is important to have transparent systems in order to be able to provide human intervention and the opportunity to remedy questionable decisions.

The above discussion reveals the global and multisectoral interest in determining ethical guidelines in this field. The search for reasonable support for this multiplicity of approaches led this investigation towards the basic sources of classical philosophy, and, with them, the language of nature. This work aims to answer the question: what are the limits in the use of AI in patient care?

An extensive review of the current state of the art in scientific literature revealed worldwide and cross-sector interest in determining ethical guidelines in this field. The search for reasonable support for this multitude of approaches led this investigation to the fundamental sources of classical philosophy and, with them, to the language of nature.

This article draws attention to important observations made by classical philosophers about the nature of human beings and technology. Key concepts in this regard are the notions of end and means. The nature of a person is to be an end in itself, while the nature of technology is to be a means. Understanding this distinction is a way of listening to the language of nature.

Plato warned that, while technology collaborates with human beings, it also offers them the opportunity to shirk their responsibility. If human beings overvalue the effectiveness of technology, they lose sight of the worth of their own agency. Contemporary authors like Floridi have worked on interesting and practical proposals to ensure that the use of advanced technologies does not undermine human agency.

In the field of healthcare and based on the causal analysis of human action, Aristotle clearly points out the different nature of the end, namely the patient’s health, in relation to the nature of the means, namely the medical art and the technology at its service.

The same distinction between what is naturally an end and what is naturally a means is reflected in another important contribution from Aristotle’s philosophy of action. Aristotle distinguishes between *praxis* and *poiesis*. Actions that are performed as an end in themselves are *praxis* and are often considered “non-productive” because they do not produce a result beyond their own realization. Productive actions (*poiesis*), on the other hand, are performed for a subsequent purpose. Human beings are capable of both types of actions. They create technology that, in turn, contributes to such productive actions. Technology, by its nature, is a product of human productive action and, in turn, it only achieves a productive, poietic movement by its nature. Global experience proves that technological instruments greatly increase the efficiency of productive activity: better, faster, and cheaper results.

However, the human being, in their practical agency, is irreplaceable. There are actions in patient care whose end is the action itself (*praxis*). These actions are constitutive of the doctor–patient personal relationship. With a high level of technological development, these actions could be emulated to some extent by AI since they still maintain a certain productive dimension. However, due to their nature (“being an end in themselves”), they cannot be replaced by a non-human agent. The nature of human action, which is an end in itself, cannot be carried out unless it is by a human agent who aims to perform the action itself. Technology is unable to perform human action, which is solely *praxis*.

## 5. Conclusions

The limits of technologies in patient care, examined in light of the language of nature (universally accepted principles), lead to the rejection of arbitrary limits imposed by authority or consensus.

Through Aristotelian analysis of action, the factual limits are recognized, whereby only the human agent is the primary efficient cause, and technology is solely the material cause: AI is an instrument by nature. Forgetting these limits leads to the devaluation of human agency at the expense of technology.

The Aristotelian distinction between *praxis* and *poiesis* asserts that human actions whose end is in themselves (*praxis*) cannot be substituted by AI. Among these "praxic" actions is love, inherently limitless, as is the primary goal of healthcare: the pursuit of the well-being of the patient and their caregivers.

To translate these philosophical conclusions into practical applications, a pragmatic guide for intelligent technologies in healthcare is provided. It demonstrates how the eight principles of AI ethics enrich the traditional four principles of bioethics with a fifth principle that shows the continuity between bioethics and the use of AI.

## Figures and Tables

**Figure 1 ijerph-20-06499-f001:**
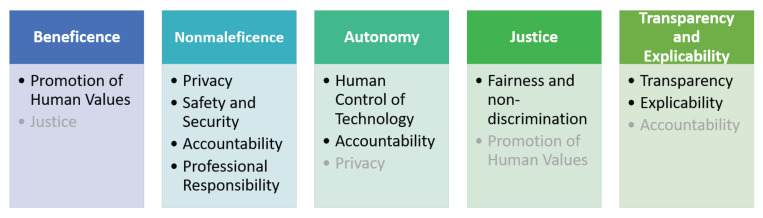
Guiding principles for healthcare intelligent technologies.

## Data Availability

Not applicable.

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
