# Peer review of "The Language of Nature and Artificial Intelligence in Patient Care"

_ijerph, 2023, doi:10.3390/ijerph20156499_

Round 1

Reviewer 1 Report

This manuscript deals with an interesting and current topic. However, it is necessary to supplement part 2 Materials and Methods with a more detailed explanation related to the methodological approach. Also, the discussion section should be designed in such a way that it leads to the conclusions that are presented in the following, namely the fifth chapter of the paper.

Reviewer 2 Report

Recently, the artificial intelligence society has arrived in us, and artificial intelligence has begun to be used in various fields of our daily lives, and it is no exception, especially in the medical field. Most AI research is focused specifically on technology, but the ethical and humanistic issues of AI cannot be ignored. This is because the humanities guide the direction of artificial intelligence technology and also set the limits of technology.

From this point of view, this work is a very fresh and valuable study However, it is regrettable that there is no claim or proof based on data, which is the main part of this journal. However, I hope that the followings will be revised to take advantage of the descriptive format and to make use of the relationship between natural language and artificial intelligence, which is the main contribution of the paper.

-Specify what is the main contribution of this study at the end of the abstract.
-It is not clear what the concepts of praxis and poiesis are in Chapter 3. Please give an easier example. In particular, please describe paragraph 3 of Section 3.3 (This exception from....) more clearly.
-The conclusion in Chapter 5 is too theoretical and general. Describe more specific conclusions.
-The contents in Chapter 6 is somewhat inconsistent with the format of this journal. Therefore, it is recommended to include them in the discussion part of Chapter 4.

Reviewer 3 Report

Dear Authors,

Thanks for giving me a chance to read this manuscript, “The Language of Nature and Artificial Intelligence in Patient Care”. The current paper shows the root of the distinction between human and artificial agency: the different nature of one and the other. Those important distinctions underpin some ethical warnings to the use of AI namely the possibility of devaluing human abilities, removing human responsibility, and eroding human self-determination.

This is an interesting and significant topic in the field of patient care. However, there are major issues in the current manuscript that should be carefully addressed to be further considered.

1.      Introduction and literature

·        This paper does not seem like a scientific paper that was supported by solid evidence, such as an experimental data report and related case summarization.

·        Please add the necessary evidence to strengthen your argument

2.      Method

·        Another big concern for me is the missing method.

·        Are any experiments or interviews associated with the arguments?

·        How do you collect data for your argument?

3.      Discussion

·        Again, the biggest concern of this study is the theoretical allocation and contribution. Authors are suggested to go through the literature and make rightful claims.

To sum up, I personally like this paper. However, the problems should be addressed in order to be further considered. Hope these suggestions help.

suggest professional proofreading 

Round 2

Reviewer 3 Report

although I still believe that the arguments should be strengthened with solid evidence, rather than deduction. The current version is better than the draft I commented last time.

although I still believe that the arguments should be strengthened with solid evidence, rather than deduction. The current version is better than the draft I commented last time.

Author Response

Thanks for your feedback. We reviewed the whole manuscript in order to strengthened our arguments and better align our deduction to the contribution of this work. We also did an English proofreading by a professional.
